# Cryo-EM structure of metazoan TRAPPIII, the multi-subunit complex that activates the GTPase Rab1

Antonio Galindo[*] , Vicente J Planelles-Herrero, Gianluca Degliesposti & Sean Munro[**]

## Abstract

The TRAPP complexes are nucleotide exchange factors that play essential roles in membrane traffic and autophagy. TRAPPII activates Rab11, and TRAPPIII activates Rab1, with the two complexes sharing a core of small subunits that affect nucleotide exchange but being distinguished by specific large subunits that are essential for activity *in vivo*. Crystal structures of core subunits have revealed the mechanism of Rab activation, but how the core and the large subunits assemble to form the complexes is unknown. We report a cryo-EM structure of the entire *Drosophila* TRAPPIII complex. The TRAPPIII-specific subunits TRAPPC8 and TRAPPC11 hold the catalytic core like a pair of tongs, with TRAPPC12 and TRAPPC13 positioned at the joint between them. TRAPPC2 and TRAPPC2L link the core to the two large arms, with the interfaces containing residues affected by disease-causing mutations. The TRAPPC8 arm is positioned such that it would contact Rab1 that is bound to the core, indicating how the arm could determine the specificity of the complex. A lower resolution structure of TRAPPII shows a similar architecture and suggests that the TRAPP complexes evolved from a single ur-TRAPP.

**Keywords** autophagy; exchange factor; Golgi complex; membrane traffic; Rab1

**Subject Categories** Membranes & Trafficking; Structural Biology

**The EMBO Journal (2021) 40: e107608**

See also: **AMN Joiner et al** (June 2021) and **BS Glick** (June 2021)

## Introduction

Small GTPases of the Rab family are major regulators of membrane traffic and organelle location in eukaryotic cells. Upon activation, they recruit to specific membranes a diverse set of effectors including molecular motors, tethering factors and regulators of both GTPases and phosphoinositides. The internal organisation of the cell thus depends on these GTPases being activated only in the correct location.

This activation is mediated by nucleotide exchange factors (GEFs) that bind the inactive GDP-bound form and catalyse the release of GDP and replacement with GTP. It has become clear that the primary determinant of the spatial accuracy of GTPase activation is the location of the relevant GEFs, and hence, understanding their structure and regulation is key to understanding the organisation of the cell (Barr, 2013; Blümer *et al*, 2013). The Transport Protein Particle (TRAPP) GEFs were discovered in yeast and subsequently found to be conserved in all known eukaryotes (Sacher *et al*, 1998; Klinger *et al*, 2013; Brunet & Sacher, 2014; Kim *et al*, 2016). In most species examined to date, including metazoans, there are two versions, TRAPPII and TRAPPIII, with TRAPPI now thought to be a subcomplex that appears *in vitro* during isolation of the other two (Choi *et al*, 2011; Brunet *et al*, 2012; Thomas *et al*, 2017). TRAPPIII activates Rab1, a master regulator of both the early secretory pathway and autophagy, while TRAPPII primarily activates Rab11, an essential player at the late Golgi where it acts in recycling from endosomes, and traffic to the surface (Jones *et al*, 2000; Wang *et al*, 2000; Cai *et al*, 2005). Rab1 and Rab11 are two of the five members of the Rab family that are present in all eukaryotes, and both are essential for the viability of all organisms so far examined (Diekmann *et al*, 2011; Kloepper *et al*, 2012). TRAPPII also has some activity on Rab1, although the *in vivo* significance of this is unresolved (Yamasaki *et al*, 2009; Thomas & Fromme, 2016; Ke *et al*, 2020). The TRAPP complexes have also been proposed to have additional roles in various processes including tethering of COPII vesicles, meiotic cytokinesis, ciliogenesis and lipid droplet homeostasis (Cai *et al*, 2007; Robinett *et al*, 2009; Westlake *et al*, 2011; Li *et al*, 2017). Consistent with the TRAPP complexes acting in key cellular processes, mutations in many of their subunits have been found in a range of familial conditions or "TRAPPopathies", including neurodevelopmental disorders, muscular dystrophies and skeletal dysplasias (Gedeon *et al*, 1999; Matalonga *et al*, 2017; Sacher *et al*, 2019).

The two TRAPP complexes share a core of seven small subunits, one of which is present in two copies to make an octamer (Fig 1A). This core is sufficient to activate Rab1 *in vitro* (Kim *et al*, 2006; Riedel *et al*, 2017). In most species, TRAPPIII has four additional unique subunits, TRAPPC8, TRAPPC11, TRAPPC12 and TRAPPC13, with the former two being essential in metazoans for cell viability and Rab1 recruitment, indicating that the core is not sufficient to correctly

MRC Laboratory of Molecular Biology, Cambridge, UK
*Corresponding author. Tel: +44 1223 267588; E-mail: agalindo@mrc-lmb.cam.ac.uk
**Corresponding author. Tel: +44 1223 267028; E-mail: sean@mrc-lmb.cam.ac.uk

activate Rab1 *in vivo* (Kim *et al*, 2016; Lamb *et al*, 2016) (Wendler *et al*, 2010; Riedel *et al*, 2017). TRAPPII has two additional unique subunits, TRAPPC9 and TRAPPC10, that are required for Rab11 activation both *in vitro* and *in vivo* (Riedel *et al*, 2017; Thomas *et al*, 2017). Crystallographic studies of the individual core subunits and their subcomplexes have revealed that the centre of the core comprises two longin domain proteins, TRAPPC1 and TRAPPC4, consistent with longin domains being present in several other Rab GEFs (Kim *et al*, 2006; Cai *et al*, 2008; Levine *et al*, 2013). Flanking these are TRAPPC5, TRAPPC6, and two copies of TRAPPC3, all of which fold into a distinct TRAPP domain that appears to have emerged in archaea, consistent with TRAPP being a universal feature of eukaryotes (Kümmel *et al*, 2005; Zaremba-Niedzwiedzka *et al*, 2017). Rab1 binds to this core region, with a subcomplex of TRAPPC1, TRAPPC3 and TRAPPC4 being sufficient for GEF activity *in vitro* (Kim *et al*, 2006; Cai *et al*, 2008). Finally, TRAPPC2 and TRAPPC2L, two longin-like proteins, are found at the two ends of the core and are required to connect the core to the specific subunits of TRAPPII and TRAPPIII (Montpetit & Conibear, 2009; Zong *et al*, 2011).

The architectures of the entire TRAPP complexes are less well understood. Low-resolution images of budding yeast TRAPPIII obtained with negative stain EM show that Trs85, the orthologue of TRAPPC8, is attached to one end of the core via the orthologue of TRAPPC2 (Tan *et al*, 2013). However, fungal Trs85 represents just the N-terminal half of TRAPPC8 and it lacks the C-terminal 600–700 residues present in plants and metazoans. In addition, *S. cerevisiae* is distinct from most other species including many other fungi, in that it lacks the TRAPPIII subunits TRAPPC11, TRAPPC12 and TRAPPC13, even though the former is essential in both *Drosophila* and mammals, and so its TRAPPIII is simpler (Fig 1A) (Wendler *et al*, 2010; Kim *et al*, 2016; Riedel *et al*, 2017; Kalde *et al*, 2019; Rosquete *et al*, 2019). Negative stain EM images of yeast TRAPPII show that the entire complex comprises Trs120 (TRAPPC9) and Trs130 (TRAPPC10) flanking the core, with this structure then dimerising with a second copy via Trs65 that links together the ends of TRAPPC9 in one copy to TRAPPC10 in the other (Yip *et al*, 2010; Pinar *et al*, 2019). However, Trs65 seems to have evolved in the yeast lineage as a distinct variant of TRAPPC13, with some other fungi having both proteins, suggesting that this form of TRAPPII is unique to a subset of fungi (Pinar *et al*, 2019).

To obtain insight into the architecture of the metazoan TRAPP complexes, we expressed recombinant TRAPPII and TRAPPIII using the *Drosophila* subunits. Single particle cryo-EM was used to obtain a structure of the TRAPPIII. This structure resolves the uncertainty about the organisation of the subunits of the core, shows how all of the additional subunits are arranged in the complex, maps the interfaces between the core and these subunits, including residues involved in genetic disease, and reveals how these additional subunits could regulate Rab binding and hence allow the core to act on different GTPases in the two different complexes.

# Results

## Biochemical characterisation of the metazoan TRAPP complexes

In previous work, we developed a protocol to express and purify recombinant forms of the *Drosophila* TRAPP complexes (Fig 1B)

(Riedel *et al*, 2017). We reported that the purified complexes are functional, with both TRAPPII and TRAPPIII having nucleotide exchange activity towards Rab1, while only TRAPPII has detectable exchange activity on Rab11. Further characterisation of these complexes shows that they are both monodisperse and monomeric, as indicated by both multi-angle light scattering coupled with size exclusion chromatography (SEC-MALS) and interferometric scattering microscopy (iSCAT) (Fig EV1A–D).

The two largest subunits of TRAPPIII, TRAPPC8 and TRAPPC11, are essential for viability of mammalian cells, and Trs85, the yeast orthologue of TRAPPC8, has been shown to bind directly to the core via TRAPPC2 (Brunet *et al*, 2013; Tan *et al*, 2013; Taussig *et al*, 2014). However, TRAPPIII has two further subunits TRAPPC12 and TRAPPC13 whose location in the complex is unknown. Expressing TRAPPIII without the TRAPPC12 and TRAPPC13 subunits reduces its size by ~100 kDa, close to the combined weight of two subunits, indicating that their absence does not affect the binding of the other nine subunits (Fig EV1A–D). This "miniTRAPPIII" complex is still able to activate Rab1 but, like the complete complex, it has no detectable activity on Rab11 (Fig EV1E).

## Cryo-EM analysis of the TRAPPIII complex

To obtain new insights into the architecture of TRAPPIII, we applied electron microscopy (EM) to examine its structure. When examined by negative staining, TRAPPIIII particles appeared homogeneous in overall size, and of rod-like or triangular appearance (Appendix Fig S1A). The particles appeared similar in cryo-EM micrographs, and 2D class averages showed clear elements of secondary structure (Appendix Fig S1B). However, initial attempts to produce a reliable 3D reconstruction failed, and we noticed that several 2D class averages had a threefold symmetry, forming an equilateral triangle (Appendix Fig S1B). This seemed inconsistent with TRAPPIII being a monomer rather than a trimer in solution, suggesting that these symmetrical particles were due to overfitting of 2D projections. Moreover, in the case of the miniTRAPPIII, similar narrow rod and triangular particles were found in the cryo-EM micrographs, but threefold symmetrical particles rarely appeared among the 2D class averages (Appendix Fig S1C). We could identify two main 2D classes of rod-like particles, one of them that partially resembled the low-resolution structure of yeast TRAPPIII obtained by negative stain (Tan *et al*, 2013), and another similar to it but with an additional density on one of its edges. We used a 3D reconstruction map of the latter as a reference map for a 3D classification of the TRAPPIII class 2D averages. This classification resulted in three different classes. The one with the best resolution and highest number of particles corresponded to an irregular triangular shape which resembled that which we observed with negative stain (Fig 1C and Appendix Fig S1A).

We reanalysed cryo-EM images of miniTRAPPIII following a similar strategy to that used for TRAPPIII. The comparison between the TRAPPIII and miniTRAPPIII class 2D averages and 3D models revealed that the flat rod was similar, but in miniTRAPPIII there was density missing at the region of interaction between the two arms (Fig 1C and D). We concluded TRAPPC12 and TRAPPC13 were located in this region of TRAPPIII, forming one of the vertexes of the triangle (Fig 1C).

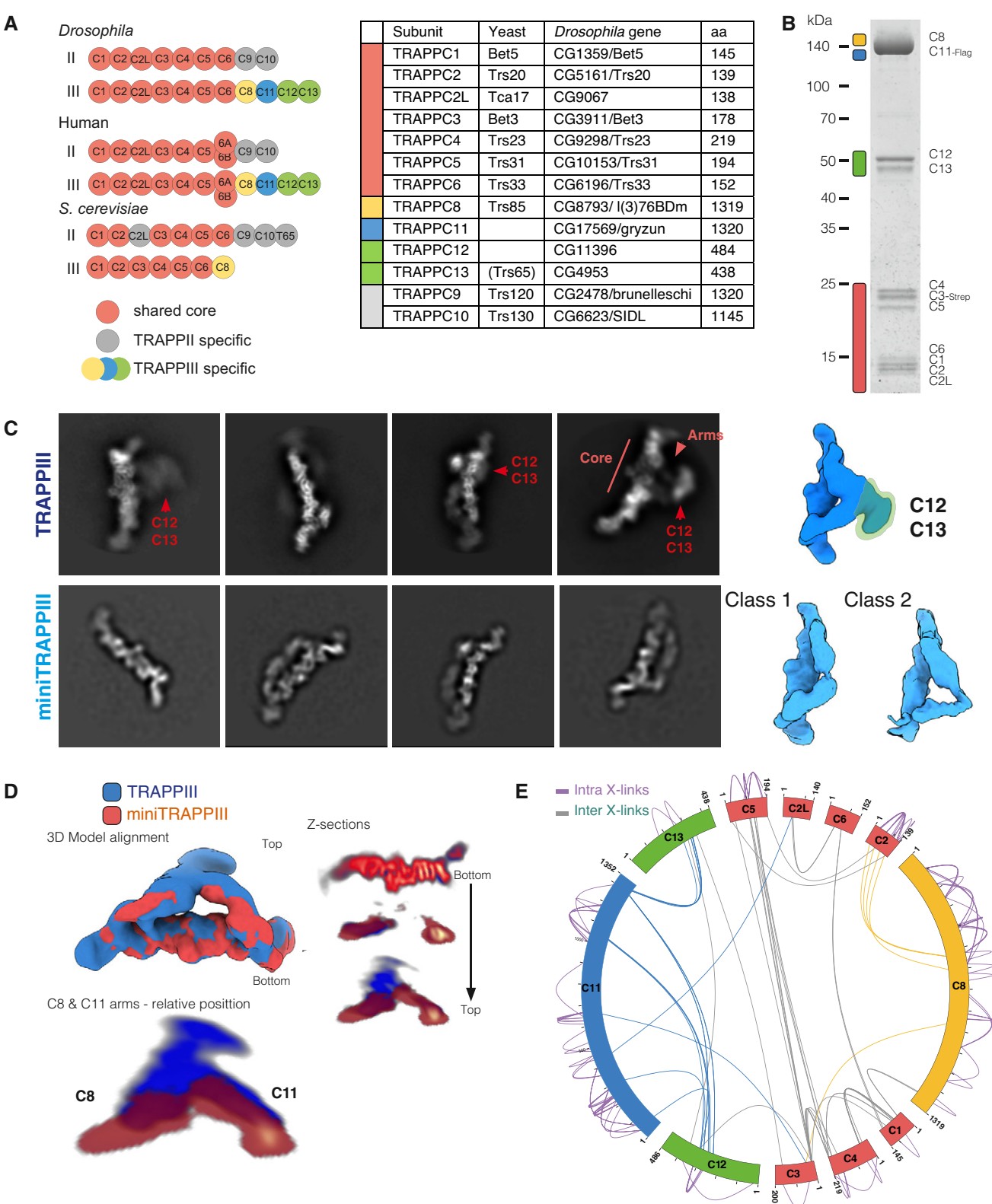

**Figure 1.**

◄

**Figure 1. Single particle imaging of the *Drosophila* TRAPPIII complex.**

A  Subunit structure of the TRAPP complexes from *Drosophila*, humans and *S. cerevisiae*. *S. cerevisiae* Trs65 is distantly related to TRAPPC13, but it appears to have a paralogue of TRAPPC13 that arose in fungi, rather than a true orthologue.

B  Coomassie blue-stained gel of recombinant *Drosophila* TRAPPIII.

C  Left: representative 2D class averages of TRAPPIII and miniTRAPPIII. Right: Low-resolution 3D models of TRAPPIII and miniTRAPPIII. Density corresponding to TRAPPC12 and TRAPPC13 is missing in miniTRAPPIII.

D  Alignment of 3D models of TRAPPIII and miniTRAPPIII. Representative Z-sections of the alignment are shown. Maximum correlation is found in the core region (bottom). A top plane from the Z-sections is enlarged. Density is missing in the absence of TRAPPC12 and TRAPPC13.

E  Circos-XL plots of the DSBU cross-links for the TRAPPIII complex (core subunits: red, specific subunits: TRAPPC8, yellow; TRAPPC11, blue; TRAPPC12 and TRAPPC13, green), inter-molecular cross-links on the outside (purple), and intra-molecular cross are on the inside (TRAPPC8 links in yellow, TRAPPC11 links in blue, others in grey).

Source data are available online for this figure.

## Refinement of TRAPPIII density map

Following averaging and refinement, the initial density map of TRAPPIII comprises an elongated flat rod with a small protrusion in the middle, and two arms that are attached at the two ends of this rod (Fig 1C). Secondary structure was better resolved inside the flat rod rather than in the two arms, which indicated two problems. Firstly, there is incomplete angular distribution of the particles due to a preferred orientation of the complex on frozen grids, (Appendix Fig S2). Secondly, the arms attached to the rod are somewhat flexible. To address the first problem, we imaged grids tilted by 19°, and combining the tilted and non-tilted datasets gave a 3D reconstruction with a nominal resolution of 5.8 Å (Appendix Fig S3A).

After several tests, we divided the reconstructed map into three different bodies for focused refinement. This approach resulted in a 4.27 Å resolution for a body containing the core subunits that form the flat rod, 4.57 Å for a second body containing one arm and the TRAPPC12 and TRAPPC13 subunits, and 5.5 Å for a third body formed by the other arm (Table 1 and Appendix Fig S3A and B).

## Arrangement of the subunits within the TRAPPIII complex

To help locate the 12 subunits of TRAPPIII within the density map, we used cross-linking mass spectrometry to identify lysine residues in proximity to each other (Fig 1E and Table EV1). As expected, there were numerous cross-links between the small subunits of the core. Of the large subunits, TRAPPC8 made several links to the TRAPPC2 subunit that is expected to be at one end of the core, as well as a link to TRAPPC3, whilst TRAPPC11 linked to TRAPPC2L and also to TRAPPC3. Association of TRAPPC8 with TRAPPC2 is consistent with what is known of yeast TRAPPIII from the effect of mutations in the TRAPPC2 orthologue, Trs20 (Brunet *et al*, 2013; Taussig *et al*, 2014). The TRAPPC12 and TRAPPC13 subunits primarily cross-linked to the C-terminal region of TRAPPC11, indicating that this part of TRAPPC11 is located in this vertex. Taken together, these results allow an unambiguous placement of the subunits within the TRAPPIII density map in which the core sits between arms formed from TRAPPC8 and TRAPPC11 with TRAPPC12 and TRAPPC13 attached at the opposite vertex. The overall shape of the complex is that of two arched arms connected at one vertex with TRAPPC12 and TRAPPC13, and then spreading apart to hold the core between their other ends like a pair of tongs (Fig 2A).

To assign the eight small subunits within the core, we used the crystal structures that have been obtained for several of these subunits from mammals, either singly or in subcomplexes comprising up to four subunits (Jang *et al*, 2002; Kim *et al*, 2006; Wang *et al*, 2014). These were used to model the seven *Drosophila* core subunits which were then built into the density map. The subunits could be readily fitted into the map with the assembly of all seven, with TRAPPC3 being present twice, creating an octamer that forms the central flattened rod (Figs 2B and C, and EV3). As expected, TRAPPC1 and TRAPPC4, which form the catalytic site to activate Rab1 (Cai *et al*, 2008), are at the centre of the rod, flanked on either side by a TRAPPC3 subunit (C3a and C3b). TRAPPC1 and TRAPPC4 share a longin domain fold, with TRAPPC4 also having a PDZ-like domain that protrudes from one side. The absence of this domain in TRAPPC1 leaves a groove on the other side of the rod, which is partially occupied by the C-terminal region of one of the two TRAPPC3 subunits (C3b). TRAPPC5 and TRAPPC2 bind one TRAPPC3 (C3b), and TRAPPC6 and TRAPPC2L bind the other (C3a). TRAPPC1, TRAPPC5 and TRAPPC2 are known to be related to TRAPPC4, TRAPPC6 and TRAPPC2L, respectively, and so the octamer has an approximate two-fold rotational symmetry, suggesting that it evolved by gene duplications adding, or altering, one half. In the octameric assembly, the greatest divergence is between TRAPPC5 and TRAPPC6. TRAPPC5 contains a disordered N-terminal region and an extra C-terminal α-helix that is not present in TRAPPC6 (Fig EV3).

## Incorporation of the TRAPP core into the rest of the complex

There are no reported crystal structures for any of the large TRAPP subunits from any species. However, in the regions of TRAPPC8 and TRAPPC11 located on either side of the core the local resolution was suitable for *de novo* model building. In the case of TRAPPC8, residues 350–660 form an α-solenoid of thirteen α-helices that includes the site of interaction with TRAPPC2 (Fig 3). The less well-resolved C-terminal region (residues 660–1,319) forms the arm that connects to TRAPPC12 and TRAPPC13 (Fig 2C). This part of TRAPPC8 is not present in the yeast orthologue Trs85, consistent with it forming an armless complex (Tan *et al*, 2013). An α-solenoid is also seen for TRAPPC11, with residues 181–566 forming fifteen α-helices that includes the interaction surface with TRAPPC2L (Fig 4). This region of TRAPPC11 has been referred to as the "foie gras domain" after the zebrafish gene in which it was first analysed as it was noted to be particularly well conserved

**Table 1. Cryo-EM data collection, refinement and validation statistics.**

|  | TRAPIII | miniTRAPIII |
|---|---|---|
| Data collection and processing |  |  |
| Magnification | 75,000× | 105,000× |
| Voltage (kV) | 300 | 300 |
| Electron exposure (e$^−$/Å$^2$) | 30 | 45.6 |
| Defocus range (μm) | −2.2/−4 | −1.5/−2.5 |
| Pixel size (Å) | 1.04 | 1.09 |
| Movies (no.) | 3671 | 3443 |
| Symmetry imposed | C1 | C1 |
| Initial particle images (no.) | 1601314 | 128478 |
| Final particle images (no.) | 353400 | 486758 |
| Map sharpening B factor (Å$^2$) | −34.32 | −45.77 |
| Map resolution (Å) | 5.8 | 4 |
| FSC threshold | 0.143 | 0.143 |
| Map resolution range (Å) | 4–9 | 3.5–7.5 |
| EMDB accession code | EMD-12056 | EMD-12063 |
| PDB accession code | 7B6R | 7B7O |

|  | Core | C8 | C11 |
|---|---|---|---|
| Refinement |  |  |  |
| Map sharpening B factor (Å$^2$) | −39.42 | −45.48 | −124.75 |
| Model resolution (Å) | 4.2 | 4.6 | 5.4 |
| FSC threshold | 0.143 | 0.143 | 0.143 |
| Model composition |  |  |  |
| Chains | 8 | 1 | 1 |
| Nonhydrogen atoms | 10,470 | 2,205 | 4,904 |
| Protein residues | 1,290 | 269 | 614 |
| Ligands | 0 | 0 | 0 |
| Protein B factors (Å$^2$) | 214.4 | 302.4 | 161.8 |
| R.m.s. deviations |  |  |  |
| Bond lengths (Å) | 0.005 | 0.005 | 0.006 |
| Bond angles (°) | 1.090 | 1.018 | 1.030 |
| EMDB accession code | EMD-12052 | EMD-12053 | EMD-12054 |
| PDB accession code | 7B6d | 7B6E | 7B6H |
| Validation |  |  |  |
| MolProbity score | 2.55 | 2.61 | 2.68 |
| Clashscore | 32.13 | 35.59 | 45.19 |
| Poor rotamers (%) | 0.43 | 0 | 0 |
| Ramachandran plot |  |  |  |
| Favoured (%) | 89.72 | 89.06 | 90.07 |
| Allowed (%) | 10.20 | 10.94 | 9.77 |
| Disallowed (%) | 0.08 | 0.00 | 0.17 |

between species (Pfam domain PF11817) (Sadler *et al,* 2005). The N-terminal part of TRAPPC11 (residues 1–180) consists of four β-strands interspersed with four α-helices (Fig 4A). The C-terminal part (residues 567–1,320) is less well resolved but forms the arm

connecting to the vertex with TRAPPC12 and TRAPPC13 (Fig 2C). Overall, we were able to model the core, and the N-terminal halves of TRAPPC8 and TRAPPC11. The C-terminal halves of these subunits, along with TRAPPC12 and TRAPPC13, were unmodelled as although helices were recognisable in many regions, the sequences could not be attributed. Nonetheless, the density map clearly shows the overall architecture of the entire complex.

This proposed architecture of TRAPPC8 and TRAPPC11 binding to the core and also linking to a vertex occupied by TRAPPC12 and TRAPPC13 is further supported by the fact that these four TRAPPIII-specific subunits are able to form a stable subcomplex when co-expressed without the core subunits (Appendix Fig S4). Finally, the structural model of the core with the two flanking solenoids can be compared with the cross-linking data. We detected 146 total cross-links, and 75 of them mapped to residues present in the structural model—32 cross-links were inter-molecular and 43 intra-molecular (Appendix Fig S5A). None of these cross-links exceeded the maximum distance constraint for disuccinimidyl dibutyric urea (DSBU) of ~30 Å, thus providing good validation for the atomic model of the TRAPPIII complex (Appendix Fig S5B and Table EV1).

### TRAPPC2 and TRAPPC2L link the core to the arms of the complex in a similar manner

The primary interactions between the core and the arms are via binding of TRAPPC2 and TRAPPC2L to TRAPPC8 and TRAPPC11, respectively. The interaction between TRAPPC2 and TRAPPC8 encloses a total surface area of ~550 Å with the TRAPPC2-binding region of TRAPPC8 formed by α-helices 9 and 11 (Fig 3A). Conserved regions in TRAPPC8 (Pro559-His567 in α-helix 9 and Trp600-Ile607 in α-helix 11) form a hydrophobic pocket needed for the interaction (Fig 3B and C). In addition, α-helix 9 contains several polar and charged residues, such as Arg562 and Lys563, that are likely to interact with key residues in TRAPPC2, including Asp46 (Fig 3C and D). Interestingly, Asp46 appears to be particularly critical for the assembly of TRAPP complexes as mutation of this residue causes spondyloepiphyseal dysplasia tarda in humans (Gedeon *et al,* 1999; Sacher *et al,* 2019) and disrupts the TRAPP complexes in yeast (Zong *et al,* 2011; Brunet *et al,* 2013; Taussig *et al,* 2014). The second conserved region is located at the beginning of α-helix 11. It contributes to binding through interaction with residues in TRAPPC2 in α-helix 1 and to a lesser extent with residues located in the loop between β-strands 1and 2 (Fig 3C and D).

The surface of interaction between TRAPPC2L and TRAPPC11 has a total area of 558 Å (Fig 4A and B). The overall arrangement of the interaction is similar to that of TRAPPC2 and TRAPPC8 (Fig EV2A). As was seen with TRAPPC2, α-helix 1 of TRAPPC2L is central to the interface and interacts with α-helices 14 and 15 of the TRAPPC11 α-solenoid (Fig 4C). We could identify two conserved regions in TRAPPC11 that are involved in the interaction. α-helix 14 (Tyr442 to Ile453) contacts TRAPPC2L between Asn33 and Lys42 (Figs 4C and D). This region is also conserved in TRAPPC2 (Fig EV2B). The other region is between residues Asp478 to Thr486 at the end of α-helix 15, where residues such as Trp484 contact TRAPPC2L α-helix 1 and the loop between β-strands 1 and 2, similar to TRAPPC8 and TRAPPC2 (Fig 4C and D).

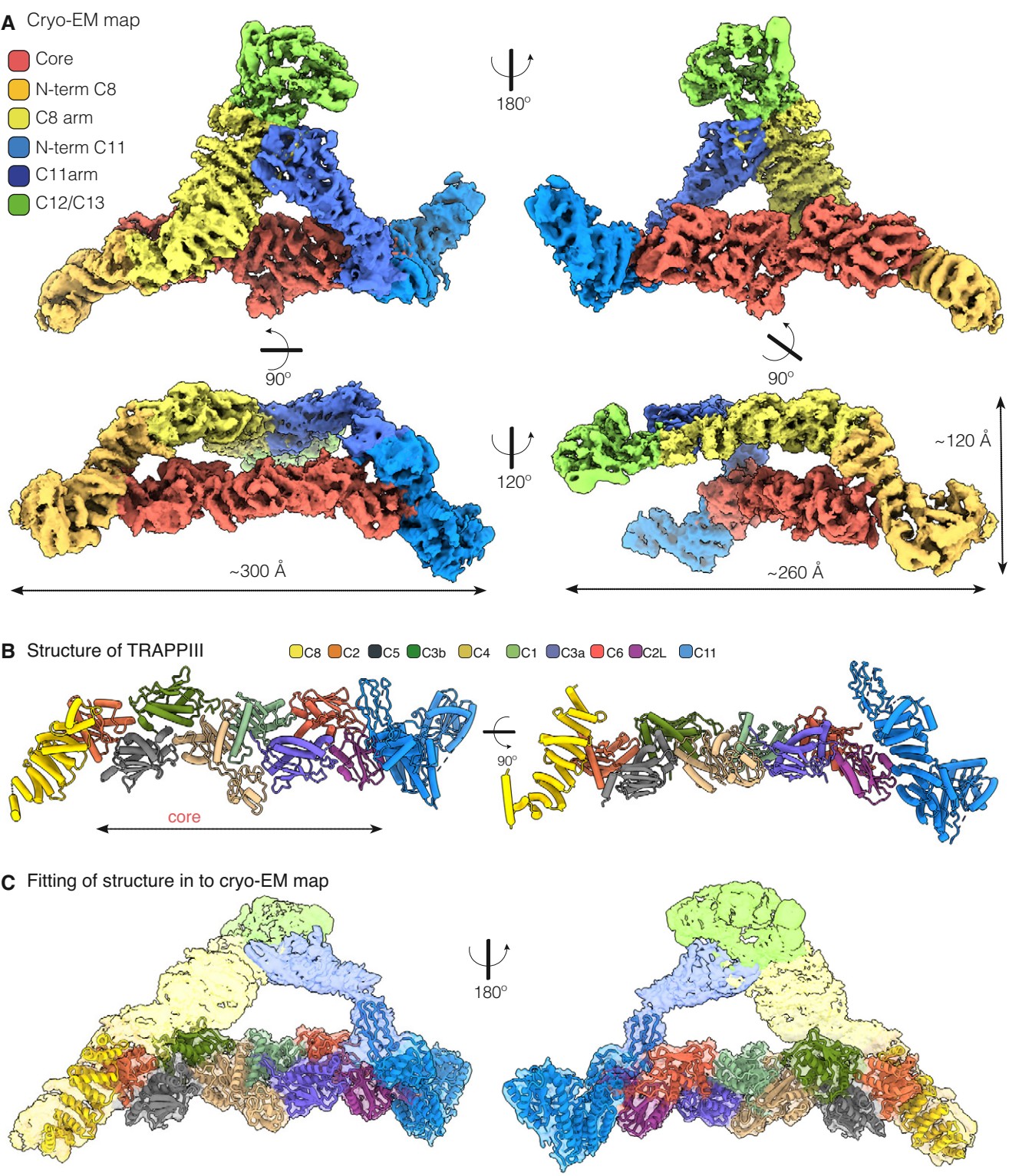

**Figure 2. Architecture of the *Drosophila* TRAPP III complexes.**

A   Cryo-EM density map coloured to show the TRAPPIII-specific subunits: TRAPPC8 (N-terminus: dark yellow; C-terminus: light yellow), TRAPPC11 (N-terminus: light blue; C-terminus: dark blue), TRAPPC12 and TRAPPC13 (light green), and the TRAPP core (red).

B   Orthogonal views of the partial TRAPPIII structural model. The subunits are depicted as pipes and planks (C1: light green, C2: orange, C2L: magenta, C3a: purple, C3b: dark green, C4: light brown, C5: grey, C6: red, C8: yellow, C11: blue).

C   TRAPPIII structural model as in (B) fitted into the cryo-EM map.

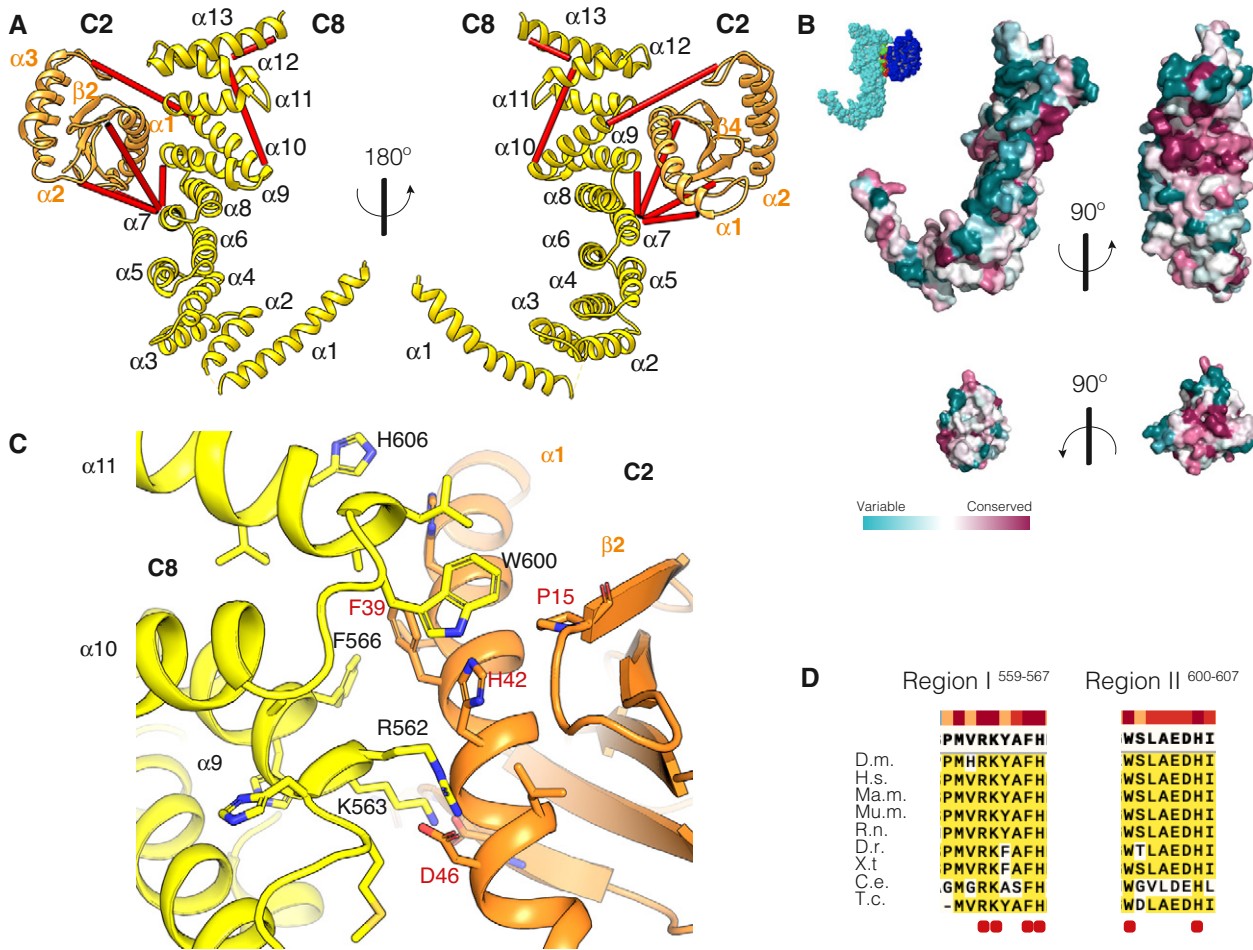

**Figure 3. The TRAPPC2-TRAPPC8 interaction surface.**

A  The TRAPPC2-TRAPPC8 subcomplex. TRAPPC8 (yellow) is formed by thirteen α-helices and binds TRAPPC2 via helices 9, 10 and 11. TRAPPC2 (orange) interacts with TRAPPC8 via α-helix 1 and the loop between β-strands 1 and 2. Cross-links mapped onto the model are shown as red lines.

B  Surface representation of TRAPPC8 and TRAPPC2 coloured according to evolutionary conservation as calculated using over orthologues identified with a Uniref 90 search with a 35% identity threshold and analysed by Consurf (Ashkenazy *et al*, 2016). Top: two views of TRAPPC8. The right one shows the surface of interaction between the two subunits. Bottom: two views of TRAPPC2. The surface of interaction is shown on the right. The inset at the top left corner shows a surface representation of the whole subcomplex for orientation purposes: TRAPPC8 is light blue, TRAPPC2 is dark blue, and the surface of interaction is coloured in red for the TRAPPC8 residues and green for the TRAPPC2 residues.

C  The TRAPPC2-TRAPPC8 interface. Structural model is coloured as in (A). Main residues involved in the interaction are shown as sticks. Labels for TRAPPC8 residues are black, for TRAPPC2 are red.

D  Alignment of the two TRAPPC8 conserved regions involved in the interaction with TRAPPC2. The residues highlighted in (C) are indicated with a red dot. Bar at the top indicates the degree of conservation.

## TRAPPC3 forms a second surface of interaction with TRAPPC8 and TRAPPC11

Two additional points of interaction between the arms and the core are visible in the TRAPPIII map as indicated by continuous densities between the flat surface of the core and the middle regions of both TRAPPC8 and TRAPPC11 (Appendix Fig S5C and D). TRAPPC8 is linked to density formed by the connection of the first two α-helices of TRAPPC3b and TRAPPC5. Similarly, density from TRAPPC11 connects to a region formed by the first two α-helices of TRAPPC3a and TRAPPC6. We could not build a model for TRAPPC8 or TRAPPC11 in these regions, but the cross-linking mass spectrometry included cross-links between TRAPPC3a Lys41 and TRAPPC11 Lys649, and between the same lysine in TRAPPC3b and TRAPPC8 Ser1259 and Thr1263 (Fig 1E and Table EV1).

## Location of the Rab1-binding site in the TRAPPIII structure

TRAPPIII activates Rab1 by catalysing the exchange of GDP for GTP and then releasing the GTP-bound Rab1 to recruit effectors to the early secretory pathway. This exchange reaction is mediated by the central subunits of the core, and a crystal structure has been obtained for these subunits from yeast in a complex with a nucleotide-free form of Ypt1, the yeast orthologue of Rab1

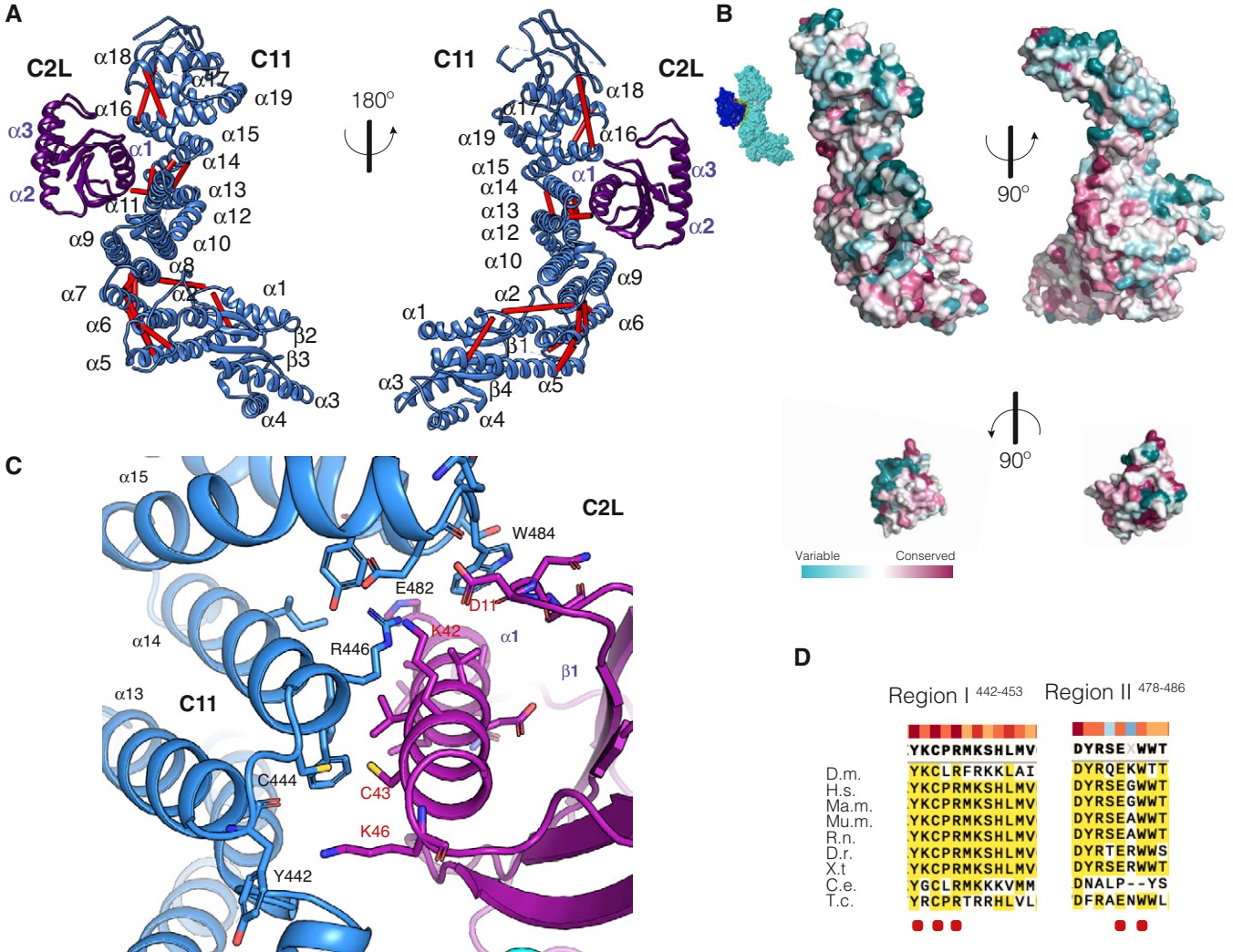

**Figure 4. The TRAPPC2L-TRAPPC11 interaction surface.**

A Two views of the TRAPPC2L-TRAPPC11 subcomplex. The TRAPPC11 built model (blue) is formed by four β-strands and four α-helices at the N-terminal region that are joined to an α-solenoid comprising α-helices 5–18. α-helices 14 and 15 form the surface of interaction with TRAPPC2L (purple), and its surface of interaction involves α-helix 1 and the loop between β-strands 1 and 2. Cross-links mapped onto the model are shown as red lines.

B Surface representation of TRAPPC11 and TRAPPC2L coloured according to evolutionary conservation (determined as for Fig 3B). Top: two views of TRAPPC11. The right one shows the surface of interaction between the two subunits. Bottom: two views of TRAPPC2L. The surface of interaction is shown on the right. The inset at the top left corner shows the whole subcomplex of TRAPPC11 (light blue) and TRAPPC2 (dark blue) with the surface of interaction coloured red for the TRAPPC11 residues and green for the TRAPPC2L residues.

C The TRAPPC2L-TRAPPC11 interface. Structural model is coloured as in (A). Main residues involved in the interaction are shown as sticks. Labels for TRAPPC11 residues are black, those for TRAPPC2L are red.

D Alignment of the two TRAPPC8 conserved regions involved in the interaction with TRAPPC2. The residues highlighted in (C) are indicated with a red dot, and the bar indicates the degree of conservation.

---

(Cai *et al*, 2008). Rab1 is highly conserved through evolution, and all of the 22 residues of Ypt1 that were found to be within 4 Å of the interface with the yeast core are identical in *Drosophila* Rab1. Likewise, the residues of the core that bind Ypt1 correspond to the most highly conserved part of the surface of the *Drosophila* core (Fig 5A). We could thus model *Drosophila* Rab1 onto the equivalent region of the *Drosophila* core and found that the GTPase fits into a space in the TRAPPIII density map (Fig 5B). Strikingly, in this position the surface of Rab1 is

precisely abutted to the arm of TRAPPC8 that arches over the core before turning away to connect to the TRAPPC12/TRAPPC13 vertex. The part of Rab1 that contacts TRAPPC8 comprises two α-helixes, α3 and α4, of the canonical Rab structure (Pylypenko *et al*, 2018) (Fig 5B). Interestingly, one of these helixes contains one of the three Rab subfamily-specific sequences (RabSF3) that were defined as being conserved between Rabs of the same family but divergent between families (Moore *et al*, 1995; Pereira-Leal & Seabra, 2000). Thus, even though RabSF3 is located away

from the switch regions that mediate binding to Rab1-specific effectors, its sequence is none-the-less specific to the Rab1 family. This suggests that contact with TRAPPC8 could stabilise the interaction between Rab1 and the core and also increase specificity. Indeed, the entire TRAPPIII complex shows significantly more exchange activity on Rab1 than does the core alone, even when the two are compared in the absence of liposomes, consistent with the presence of TRAPPC8 promoting the interaction of Rab1 with the complex (Fig 5C).

Finally, we investigated the relevance of the flexibility of TRAPPIII for Rab1 binding. As noted above, the limits on the resolution of the density map implied that the complex is not entirely rigid. The best resolved part of the map is the core and the associated regions of TRAPPC8 and TRAPPC11. To assess its movement relative to the rest of the complex, we used multi-body refinement to look at variation of the particles within the dataset and analysed the results by principle component analysis (Nakane *et al*, 2018). Almost half of the variability between particles can be accounted for by a movement vector corresponding to a rocking of the arms relative to the core (Appendix Fig S6A and B, and Movie EV1). This indicates that the arms have sufficiently flexibility for TRAPPC8 to move over the Rab1-binding site to the point that it would block binding of the GTPase to the catalytic site on the core (Appendix Fig S6C). This provides a possible mechanism by which the interactions formed by the four subunits of the arms could regulate exchange activity.

GDP-bound forms of Rabs bind the cytosolic chaperone GDP-dissociation inhibitor (GDI) that masks the C-terminal prenyl groups. This enables the Rab to be soluble in the cytosol, and activation of Rabs is believed to occur after GDI has released the GDP-bound form on to a membrane. This means that GEFs like TRAPP act on the membrane rather than in the cytosol (Pylypenko *et al*, 2006; Barr, 2013; Blümer *et al*, 2013). Consistent with this, the TRAPPs have greater activity towards Rabs bound to liposomes, which indicates that the TRAPPs interact with the membrane surface, an interaction that could be promoted *in vivo* by other proteins present on the membrane (Thomas & Fromme, 2016; Riedel *et al*, 2017). Therefore, in addition to allowing Rab1 to access the catalytic site in the core, the structure of TRAPP needs to be compatible with the substrate Rab1 being connected to a lipid bilayer via the unstructured hypervariable domain that links the GTPase to the prenyl groups that mediate membrane attachment (Li *et al*, 2014). The location of the hypervariable domain when Rab1 is bound to TRAPP is unknown as it was not included in the form of Rab1 used to generate a crystal structure with the core subunits. However, modelling the TRAPPIII structure on a flat surface places the Rab1-binding site 55 Å above this surface (Fig 6A), a distance that would be readily accommodated by the ~95 Å that the 27 residue hypervariable domain of *Drosophila* Rab1 (Gly177-Gly203) could reach at its maximum extent. It should be stressed that this orientation on the surface is hypothetical, based on the assumption that the vertexes of the complex serve to mediate membrane contact. It is thus formally possible that the complex is positioned perpendicular to the membrane, which would move the Rab-binding site closer, but we can at least say that earlier proposals that the sides or the underneath of the core could contact the membrane are not sterically possible as the vertices extend beyond the core (Kim *et al*, 2006; Cai *et al*, 2008),

whereas having all three vertices on the membrane would be compatible with exchange activity.

### Architecture of the TRAPPII complex

The TRAPPII complex shares the core subunits with TRAPPIII but has different additional subunits which allow it to activate Rab11 (Fig 1A). To compare the overall architecture of the two complexes, we expressed a recombinant form of *Drosophila* TRAPPII and subjected it to single particle cryo-EM imaging. A low-resolution 3D map shows that the overall architecture of TRAPPII is similar to that of TRAPPIII, with an elongated rod of the dimensions of the core attached to two arms that connect at their opposite ends to form an irregular triangle (Fig EV4A). Application of cross-linking mass spectrometry indicates that TRAPPC9 is linked to the core through TRAPPC2, and TRAPPC10 is linked via TRAPPC2L, with the latter showing a cross-link via the same Lys47 residue that linked to TRAPPC11 in TRAPPIII (Fig EV4B and C, Table EV1). This is consistent with studies in yeast where the TRAPPC2L orthologue Tca17 is required for association of Trs130 with the TRAPPII complex (Choi *et al*, 2011; Milev *et al*, 2018). The pattern of cross-linking between the core subunits is similar to that found in TRAPPIII, and TRAPPC3 Lys 41 and TRAPPC6 Lys 104 also link to TRAPPC10, analogous to the links these core subunits form to TRAPPC11. Together, these findings show that both metazoan TRAPP complexes share an architecture that consists of a central core held between two elongated arms.

## Discussion

The TRAPP complexes have emerged as arguably the two most critical activators of Golgi Rab function, with Rab1 acting as the master regulator of entry into the early compartments of the stack, as well as being a key player in autophagy. Biochemical and structural studies have elegantly shown that Rab activation *in vitro* requires only three of the small subunits at the core of these large structures (Kim *et al*, 2006; Cai *et al*, 2008). The presence of further, and larger, subunits presumably reflects the need for precise temporal and spatial control of the activation of these essential GTPases, but how these subunits might exert control over the core GEF activity has been unclear. The cryo-EM structure of the entire TRAPPIII complex presented here clearly shows how the arms of the complex are attached to the core and resolves long-standing uncertainty on this issue. Previous structural and genetic studies have provided unambiguous evidence that TRAPPC8 binds to TRAPPC2 which is present at one end of the core (Kim *et al*, 2006; Brunet *et al*, 2013; Tan *et al*, 2013; Pinar *et al*, 2019). Likewise, it is clear that in TRAPPII, TRAPPC9 binds to the same subunit. However, the situation for the other arms has been less clear as TRAPPC2L is absent from yeast TRAPPIII, and there is no crystal structure of a complex between TRAPPC2L and other core subunits. It has been proposed that TRAPPC2L acts in TRAPPII in yeast to attach TRAPPC11 to the core, but this has not been universally accepted (Montpetit & Conibear, 2009; Choi *et al*, 2011; Lipatova & Segev, 2019). Our results unambiguously place TRAPPC2L at the opposite end of the core from TRAPPC2 and show how it attaches to TRAPPC11. TRAPPC2L is likely to have the equivalent position in TRAPPII so as to attach

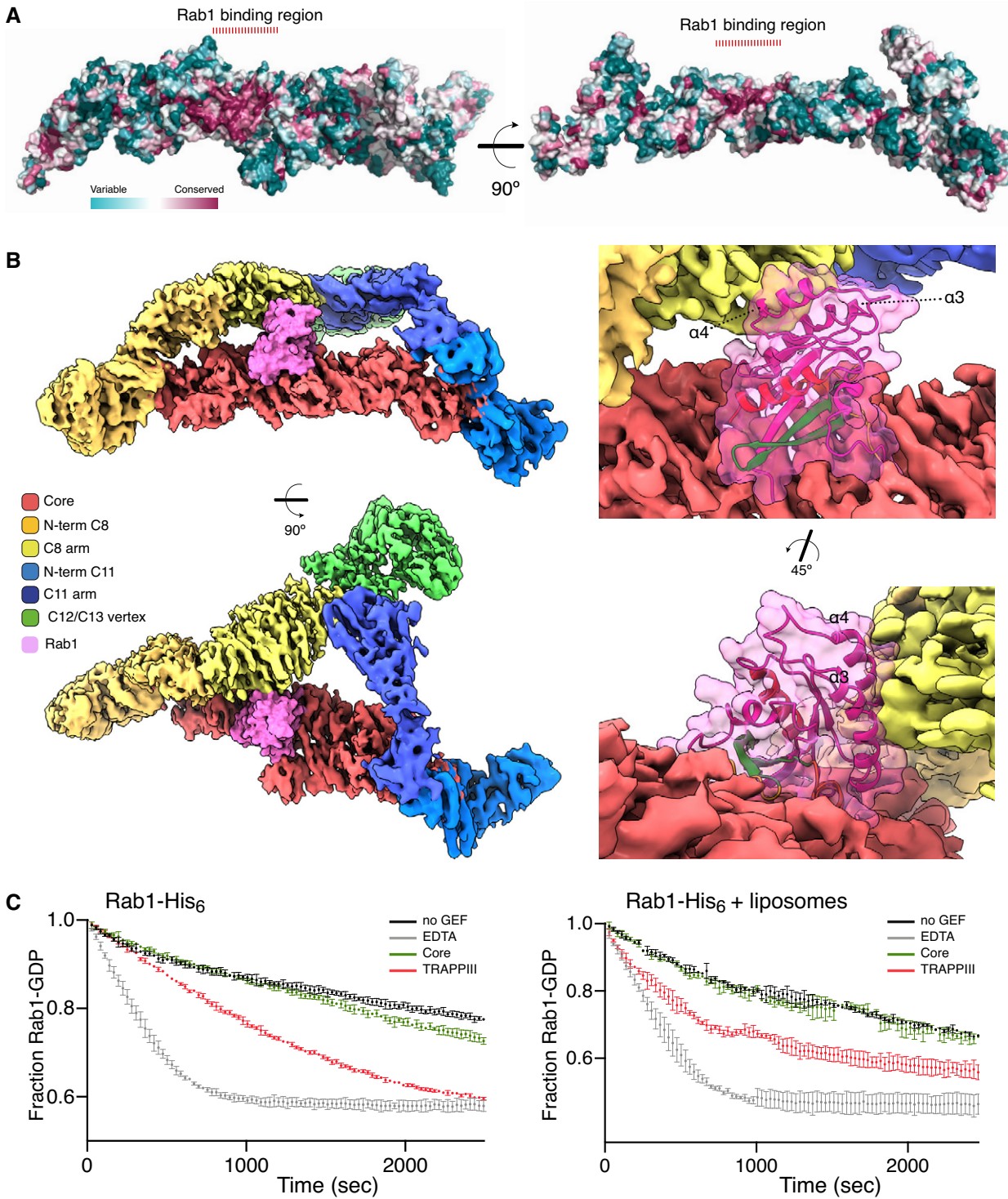

**Figure 5. Rab1-binding site in TRAPPIII.**

A  Left: TRAPPIII model surface coloured by evolutionary conservation (determined as in Fig 3B). A highly conserved region on one surface of the core corresponds to the Ypt1-binding region in the complex with the central subunits of the yeast TRAPP core (Cai *et al*, 2008).

B  Density map of *Drosophila* TRAPPIII with Rab1 (pink) modelled based on the location of Ypt1 bound to the core of yeast TRAPP. When bound to the core, Rab1 abuts the arm of TRAPPC8 (yellow). Enlargements with the Rab1 ribbon structures showing that the canonical Rab helices α3 and α4 face the surface of TRAPPC8.

C  Release of mant-GDP from Rab1-His$_6$ (250 nM) in the absence or presence of synthetic liposomes. Rab1 was loaded with mant-GDP, and fluorescence measured following addition of GTP, either alone (black) or with 25 nM GEF (entire TRAPPIII, red; core, green) or with 10 mM EDTA (grey). In both cases, TRAPPIII increases the rate of nucleotide exchange more than the core. Mean and SEM from three experiments.

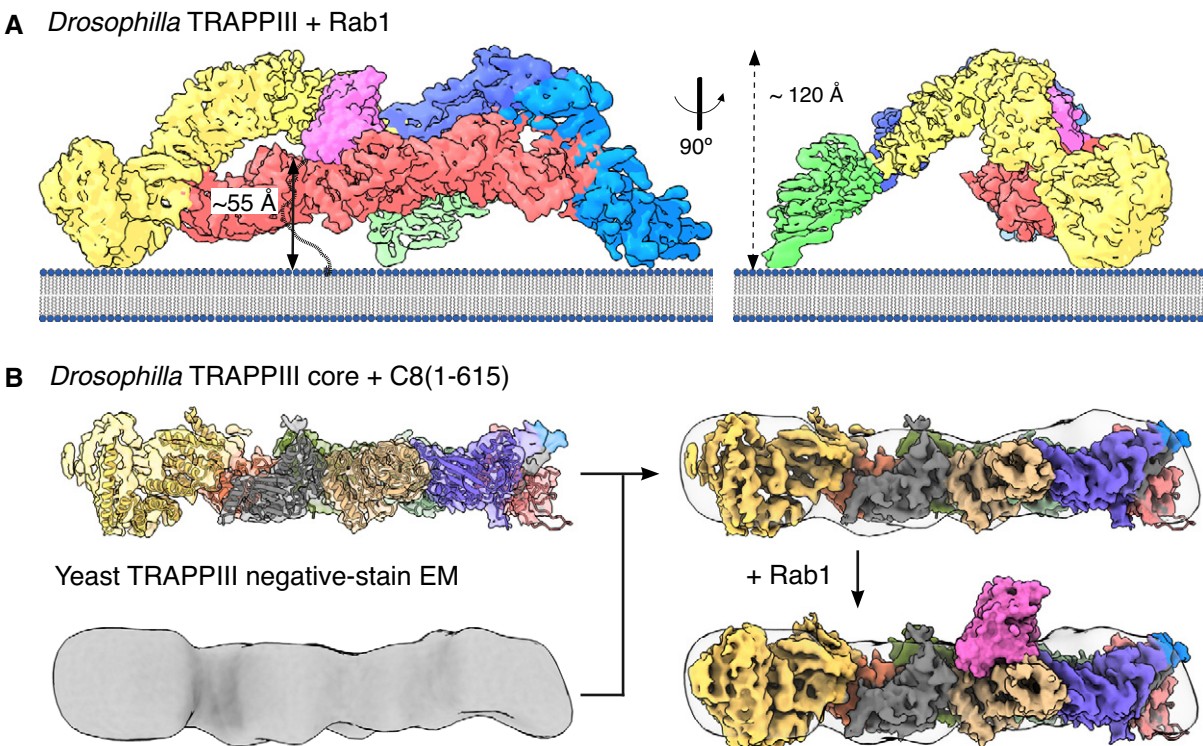

**Figure 6. Comparison of metazoan and yeast TRAPPIII.**

A  Model of TRAPPIII with bound Rab1 on a flat lipid membrane. The distance between the surface of the Rab1-binding site and the membrane is 55 Å, shorter than the predicted maximum length of the unstructured hypervariable domain of Rab1 that connects it to its C-terminal lipid anchor (shown as hashed line). This location on the bilayer is hypothetical based on the assumption that the vertexes of the complex are involved in membrane recruitment. In addition, on a curved bilayer, such as a vesicle, the distance to the membrane would be shorter.

B  Density map, with fitted structure, for the TRAPPIII core plus the N-terminal half of TRAPPC8 fitted into the previously reported negative stain density map for *S. cerevisiae* TRAPPIII (Tan *et al*, 2013). The N-terminal half of TRAPPC8 corresponds to the shorter orthologue, Trs85, present in yeast. Rab1 is also modelled to show that, unlike the case for the *Drosophila* TRAPPIII, the yeast orthologue is not in a position to contact the shorter version of TRAPPC8.

TRAPPC10. Interestingly, in *Drosophila* or fungal mutants lacking TRAPPC10, TRAPPC2L appears to dissociate from the core, suggesting its binding is stabilised by the interaction, possibly by virtue of the triangulation to the other end of the core via the other arm (Riedel *et al*, 2017; Pinar *et al*, 2019). Alternatively, quality control mechanisms in the cell may recognise the partly assembled complex and degrade some of the subunits. The fact that the related TRAPP2C and TRAPPC2L subunits have equivalent roles at opposite ends of the complex is echoed by the fact that the sequence of TRAPPC11 is distantly related to that of TRAPPC10, with the same being true for TRAPPC8 and TRAPPC9 that bind TRAPPC2 (Wendler *et al*, 2010; Scrivens *et al*, 2011). This indicates that a eukaryotic precursor had a single ur-TRAPP, and gene duplication gave rise to the two TRAPPs that appear to have been present in the last eukaryotic common ancestor.

The TRAPPIII structure also reveals the location of TRAPPC12 and TRAPPC13 in the complex, showing that they are present at the joint between the TRAPPC8 and TRAPPC11 arms. Unlike, the arms themselves, these subunits do not seem to have equivalents in *Drosophila* TRAPPII. The TRAPPII complex of budding yeast has an additional subunit, Trs65, that was originally proposed to be a yeast

homolog of TRAPPC13, but now appears to be a relative that arose by duplication in fungi, with some budding yeast then losing TRAPPC13 itself, along with TRAPPC12 and TRAPPC11 (Choi *et al*, 2011; Riedel *et al*, 2017; Pinar *et al*, 2019). Interestingly, something similar seems to have happened in some other phyla with a TRAPPC13 relative being recently found to associate with at least a subset of TRAPPII in plants (TRIPP), and vertebrates (C7orf43/ TRAPPC14), suggesting that the vertex of the TRAPP arms is a convenient place to bolt on additional subunits (Cuenca *et al*, 2019; Garcia *et al*, 2020).

Modelling of Rab1 into the complex, and comparison of the GEF activity of the whole complex with the core, both suggest that Rab1 contacts the TRAPPC8 arm positioned above the active site. This part of TRAPPC8 is present in most species from humans to plants and protozoa, but has been lost in some fungi. Thus, in *S. cerevisiae*, Trs85 corresponds to the first ~650 residues of the 1,319 residue *Drosophila* TRAPPC8. This presumably reflects there being no TRAPPC11, TRAPPC12 and TRAPPC13 to connect to. Indeed, the *Drosophila* TRAPPIII core plus the first half of TRAPPC8 fits closely into the overall shape of *S. cerevisiae* TRAPPIII determined by negative stain EM (Fig 6B) (Tan

*et al*, 2013). The TRAPP core in yeast is sufficient to activate the Rab1 orthologue Ypt1 in solution, with the presence of Trs85 increasing the activity towards the GTPase when bound to membranes, presumably by promoting membrane recruitment of the complex rather than via direct binding of Trs85 to Ypt1 (Thomas *et al*, 2017).

A key question that remains is the role of the rest of the large arms of the TRAPP complex apart from the potential binding to the substrate GTPase. The structure of TRAPPIII reveals how the arms could serve as regulators despite being attached via subunits that are distal to those that mediate GEF activity. Like other GEFs, TRAP-PIII is likely to be regulated by recruitment to the membranes where it can access GDP-bound Rab1 (Barr, 2013; Blümer *et al*, 2013). The size of the arms provides a large surface area that components of membrane traffic or autophagy could bind to without sterically inhibiting exchange activity, and indeed, interactions have been reported between TRAPPs and a wide range of potential regulators including the Sec23/Sec24 and the Sec13/31 subunits of the COPII coat, the COPI coat, Arf1 exchange factors, the Rab GAP TBC1D14 and the autophagy proteins Atg2 and Atg9 (Kakuta *et al*, 2012; Tan *et al*, 2013; Lamb *et al*, 2016; Stanga *et al*, 2019). In addition, yeast TRAPPII can be activated *in vitro* by the small GTPase Arf1 (Thomas *et al*, 2018). However, the architecture of the TRAPPIII complex indicates that the arms could also have more direct effects on activation. Firstly, TRAPPC8 is positioned such that it would contact Rab1 bound to the GEF active site on the core, and this could both enhance the rate of exchange and also improve selectivity for Rab1 over other Rabs. Secondly, the flexibility of the arms is such that TRAPPC8 could move so as to interfere with, rather than augment, access to the active site, and therefore interactions that moved the arms could alter the activity of membrane-bound TRAPPIII. Finally, we observe apparent contacts between both arms and the TRAPPC3 subunits near the centre of the core which raises the possibility of allosteric regulation. Clearly, further work will be required to address the *in vivo* significance of these various possible modes of regulation, but hopefully the architecture reported here will greatly facilitate this by guiding the construction of specific alterations to the complex.

## Materials and Methods

### Expression and purification of TRAPP complexes

The TRAPP purification protocol is based on previous work (Riedel *et al*, 2017). The complexes were expressed in insect cells (Sf9 or Hi5 lines) using the MultiBac System (Nie *et al*, 2014). A pACEBac1 plasmid containing the seven *Drosophila melanogaster* core subunits, pACEBac1-C1-C6, was fused using Cre recombinase (New England Biolabs), to a pIDS vector containing TRAPPC9 and TRAPPC10 to generate the plasmid pACEBac1-TRAPPII-complete. A similar strategy was followed to construct the pACEBac1-TRAPPIII-complete: TRAPPC8 and TRAPPC11 were cloned into pIDS, and TRAPPC12 and TRAPPC13 were cloned into pIDC. The resulting plasmids were recombined into pACEBac1-C1-C6 vector to express miniTRAPPIII, with only TRAPPC8 and TRAPPC11, or the complete TRAPPIII (Riedel *et al*, 2017). The TRAPPC3 subunit was tagged with Strep-TagII, and TRAPPC11 (TRAPPIII) or TRAPPC10

(TRAPPII) was FLAG-tagged, both at the N-terminus. An additional pACEBac1-C1-C6 with the TRAPPC2L subunit tagged with the ZZ domain at the N-terminus was used for the expression of the TRAPP core. The linker sequence between each subunit and the tag included a site for the HRV-3C protease. The pIDS and pIDC plasmids containing the specific TRAPP subunits were also fused to an empty pACEBac1 to express these subunits in the absence of the TRAPP core subunits.

Bacmids were made using the EMBacY system (Nie *et al*, 2014). A 500 ml suspension of Sf9 cells ($2 \times 10^6$ cells/ml) was infected with 5 ml of fresh P2 baculovirus and incubated at 27°C and 124 rpm. Cells co-expressing the TRAPP core, TRAPPII or TRAPPIII were harvested after 66 h (at 75–80% viability) by centrifugation at $2,250 \times g$ for 10 min at 4°C. Pellet was washed once with PBS, centrifuged again and processed immediately for the whole TRAPP complexes, or kept at −80°C in the case of the TRAPP core. Initially, pellets were resuspended in Buffer A (50 mM HEPES-KOH pH 7.44, 150 mM KAc, 1 mM DTT, 0.1% IGEPAL CA-630) with inhibitors (1 mM PMFS, cOmpleteTM, 0.4 µM pepstatin, 0.24 µM leupeptin, 5 µM MG132) at a ratio of 30 ml per 500 ml of initial culture. The cell suspension was vortexed and incubated at 4°C for 10 min, before lysis by 15–20 strokes of a tight-fitting dounce homogeniser. The lysate was clarified by centrifugation at $32,000 \times g$ for 30 min at 4°C. Cleared lysate was mixed with the appropriate equilibrated slurry: Strep-Tactin Superflow Plus (Qiagen) (400 µl per 500 ml of initial culture), Anti-FLAG M2 affinity gel (Sigma, A2220) (100 µl per 500 ml culture) or IgG Sepharose (6 Fast Flow, GE Healthcare) (500 µl per 500 ml culture), and incubated on rotation wheel for one hour at 4°C. Beads were washed three times with ten bead volumes of Buffer A plus 0.05% IGEPAL CA-630. Bound material was eluted by washing the beads with five bead volumes of Buffer A containing either 100 µg/ml FLAG peptide (anti-FLAG) or 2.5 mM desthiobiotin (Strep-Tactin). The eluted fraction was analysed by SDS–PAGE, concentrated and buffer exchanged. Alternatively, the bound complexes were eluted by tag cleavage incubating the slurry with PreScission protease (~10 U/ml) overnight at 4°C. The eluted solution was mixed with glutathione–Sepharose to remove the PreScission protease.

The TRAPP core complex was purified further by gel filtration (SEC) using Superose 10/30 (GE Healthcare) (Appendix Fig S4) for small samples, or Superdex200 16/100 equilibrated in Buffer A plus 0.005% IGEPAL CA-630. This protocol was escalated to 6 l cultures (12 × 500 ml) for TRAPPII, TRAPPIII and miniTRAPPIII. We found that increasing the KAc concentration prevents the formation of aggregates during SEC step purification and so the composition of Buffer A composition was adjusted to 250 mM KAc, and the IGEPAL CA-630 removed during the subsequent bead washing. Detergent-free samples were concentrated up to 3–5 mg/ml, and ~100 µl fractions were loaded onto a TSKgel G4000SWXL column (TOHO Bioscience) in 50 mM HEPES-KOH pH 7.44, 250 mM KAc, 1mM DTT (Buffer B). Eluted peaks were collected in 100 µl fractions and analysed by SDS–PAGE and Coomassie staining (Fig EV1A). Protein identification by mass spectrometry was used to assess the integrity of the purified complexes. The best yield for the purification of whole TRAPP complexes was obtained using the TRAPPC10 or TRAPPC11 FLAG-tagged subunits as baits for the affinity chromatography. There was no difference in stoichiometry or *in vitro* GEF activity between complexes obtained by FLAG peptide elution

or by HRV-3C protease cleavage, and so we continued with the former method.

### SEC-MALs and iSCAT analysis

For SEC-MALs, purified TRAPPIII and miniTRAPPIII (~100 µl at 0.5 mg/ml) were resolved on a Superose 6 10/300 column (GE Healthcare) in Buffer B, with a flow rate of 0.5 ml/min. Protein was detected with 280 nm UV light (Agilent Technology 1260), a quasielastic light scattering module (DAWN-8+, Wyatt Technology), and a differential refractometer (Optilab T-rEX, Wyatt Technology). Molar masses of peaks in the elution profile were calculated from the light scattering and protein concentration, quantified using the differential refractive index of the peak, assuming dn/dc = 0.186, with ASTRA7 (Wyatt Technology).

For iSCAT, TRAPPIII, TRAPPII and miniTRAPPIII were diluted to a final concentration of 25, 50 and 100 nM irrespectively in Buffer B. 10 µl of each sample was applied to 10 µl of Buffer B on a cleaned glass coverslip (no. 1.5, 24 × 50 mm) coated with a silicone well frame and analysed for 10 min at a rate of 600 frames/ min with an ONEMP mass photometer (Refeyn LTD, Oxford, UK). 25 nM and 50 nM BSA solution were used as standards for calibration. For each recording of a BSA standard, a histogram was made and fitted with Gaussians according to how many peaks are resolved. Fitted centres of these Gaussians and the corresponding masses that they are assigned to were plotted and fitted to a straight line. The resulting parameters were used as conversion between measured contrast and mass for the TRAPP samples (Cole *et al*, 2017). Data were acquired and analysed using AcquireMP and DiscoverMP (v1.2.3) (Refeyn LTD, v1.1.3). Measurements were repeated at 4°C and room temperature with similar results.

### Negative stain EM

After gel filtration, TRAPPIII samples were diluted to 0.008–0.009 mg/ml (~16–20 µM) and applied to EM grids. 3 µl of diluted sample was deposited onto a glow-discharged (Edwards S150B, 30 s, 30–50 mA, 1.2 kV, 10–2 mbar) continuous carbon grid (CF400-CU-UL, Electron Microscopy Sciences). After one minute at RT, the grid was blotted and washed by immersion in a 100 µl drop of fresh 2% uranyl acetate and blotted again. Then, the grid was stained by two rounds of 2% uranyl acetate immersion for 30 s and blotting, before being air-dried. Micrographs were collected on a Tecnai T12 microscope (Thermo Fisher Scientific) operating at 120 keV with a tungsten electron source and a 2k × 2k CCD camera (Orius SC200W, Gatan, Inc.). Nominal magnification was 15,000×, giving a 3.50 Å/pixel sampling at the object level. Images were collected with a dose of 50 e⁻/Å² and a nominal defocus of −1 µm. In total, 100 micrographs were collected. TRAPPIII particles were manually picked and subjected to initial 2D classification using Relion 3.0 (Zivanov *et al*, 2018). Automated particle picking was made using EMAN2, and particle coordinates were imported into Relion 3.0. The initial 22,986 particles were subjected to two rounds of Class2D classification resulting in 25 class averages. 3,123 particles were sorted to build a 3D initial model *de novo*. This model was used as a reference map for 3D refinement of the total subset of 10,077 good quality particles. The final model was obtained after Class3D classification and another round of 3D refinement.

### Cryo-EM grid preparation

After gel filtration, TRAPPIII, TRAPPII or miniTRAPPIII was diluted to 0.9–1 mM (0.5 mg/ml) in buffer supplemented with IGEPAL CA-630 to reach a final concentration of 0.005%. Samples were applied to freshly glow-discharged (Edwards S150B, 45 s, 30–50 mA, 1.2 kV, 10–2 mbar) copper holey carbon grids (Quantifoil, Cu-R1.2/1.3) under 100% humidity. Excess sample was blotted away, and the grids were subsequently plunge-frozen in liquid ethane using a Vitrobot Mark III (Thermo Fisher Scientific).

### Data collection

TRAPPIII: A total of 3,671 movies were recorded on a Titan Krios electron microscope (Thermo Fisher Scientific-FEI) operating at 300 kW with a calibrated magnification of 75000x and corresponding to a magnified pixel size of 1.04 Å. Micrographs were recorded using a Falcon III direct electron detector in counting mode with a dose rate of ~0.5 e/Å2/s and defocus ranging from −2.2 µm to −4 µm. The total exposure time was 60 s, and intermediate frames were recorded in 0.8-s intervals, resulting in an accumulated dose of ~30 e/Å2 and a total of 75 frames per micrographs. 1,190 movies out of the 3,671 data sets were collected with the stage titled at 19°, this angle chosen according to the output from the cryoEF algorithm (Naydenova & Russo, 2017).

MiniTRAPPIII: A small data set of 385 micrographs was acquired under the same conditions described for TRAPPIII. Data derived from these micrographs were used for building a partial *ab initio* 3D model used as a reference map for TRAPPIII and miniTRAPPIII.

A second data set of 3,443 micrographs was acquired on a Titan Krios EM operating at 300 kW with a calibrated magnification of 105,000× and corresponding to a magnified pixel size of 1.047 Å. Micrographs were recorded using a K2 direct electron detector (Gatan) equipped with a Cs corrector and an energy filter. Images were collected over 12 s in counting mode with 0.3 s (~e⁻/Å²/s) frame time and a slit width of 20 eV. The total exposure was 45.6 e/Å², and the defocus ranged from −1.5 µm to −2.5 µm.

TRAPPII: A total of 364 micrographs were recorded on a Titan Krios III EM operating at 300 kW with a calibrated magnification of 75,000× and corresponding to a magnified pixel size of 1.09 Å. Settings for the acquisition were similar to those for TRAPPIII (Falcon III in counting mode, ~0.5 e⁻/Å²/s, defocus −2.2 µm to −4 µm, exposure 60s, total dose ~30 e⁻/Å²).

### Image processing

Dose fractionated image stacks were subjected to beam-induced motion correction and filtered according to the exposure dose using MotionCor2 (Zheng *et al*, 2017). The sum of each movie was applied to CTF parameters determination by CTFFIND 4.1 (Rohou & Grigorieff, 2015). For the tilted data, the CTF was corrected according to the focus gradient of each particle using goCTF (Su, 2019). A custom script was written to run the goCTF v1.2.0 software in batch. Particles were picked using cryOLO 1.5 (Wagner *et al*, 2019). Particles from the small miniTRAPPIII data set were subjected to 2D classification, and 22,897 particles were chosen to create an *ab initio* 3D model using the Frealign tool implemented in cisTEM (Grigorieff, 2016; Grant *et al*, 2018). This resulted in a rough 3D map at 7 Å

used for 3D classification of particles from the larger TRAPPIII and the miniTRAPPIII data sets. In the case of TRAPPIII, tilted and non-tilted data were CTF corrected, extracted and subjected to two rounds of reference-free 2D classification using Relion 3.1. Selected 2D classes were used for a 3D classification resulting in three different classes that were similar among the different data sets. The corresponding particles to the cleanest 3D class from each data set were joined, reextracted and subjected to an additional round of 2D classification. The selected particles after this round were 3D refined. After this, 353,400 particles were subjected to 3D masked refinement followed by map sharpening in Relion 3.1. The estimated CTF parameters were refined, and per-particle reference-based beam-induced motion correction was performed using Bayesian polishing. The final map has a global resolution of 5.8 Å. Reported resolution is based on the gold-standard Fourier shell correlation (FSC) using the 0.143 criteria. Local resolution was estimated using the Relion 3.1 implementation. A similar strategy was followed for miniTRAPPIII, but the global resolution was higher than the consensus map for TRAPPIII, at 4 Å, but with a higher range in the local resolution. This is due to the higher number of micrographs and the strong preferential orientation of this complex (EOD 0.62; Naydenova & Russo, 2017) vIn the case of TRAPPII; 43,161 particles were picked using crYOLO. After several runs of 2D classification, 22,570 particles were selected to generate a rough *ab initio* 3D model. This model was used as a reference map for a 3D classification. Particles corresponding to the best 3D classes were joined and subjected to 2D classification, and 3084 good particles were selected to generate a 3D model at 15 Å resolution.

## Multi-body refinement

To improve the density, increase the resolution and characterise the conformational dynamics, we performed multi-body refinement with RELION 3.1 (Nakane *et al*, 2018). TRAPPIII was divided into three or four discrete bodies composed initially by the whole flat rod, the TRAPPC11 arms and the TRAPPC8 arms plus the TRAPPC12-TRAPPC13 vertex, with the latter being isolated as an additional body for the four bodies approach. In later trials, the core alone constituted one body, and the whole TRAPPC11 and the whole TRAPPC8 plus TRAPPC12 and TRAPPC13 the other two. Masks for multi-body refinement were made in UCSF Chimera 1.15 from the consensus map (Pettersen *et al*, 2004). The standard deviation of the Gaussian prior on the rotations was set to 10 degrees for all three bodies. The standard deviations on the body translations were all set to two pixels. The maps for the three discrete bodies after multi-body refinement were post-processed individually and combined using Phenix (Liebschner *et al*, 2019). There was an increase in resolution (Appendix Fig S3, body 1: core 4.2 Å, body 2: C8-C12-C13 4.4 Å and body 3: C11 5.5 Å), enabling interpretation of the density for the N-terminal regions of TRAPPC8 and TRAPPC11.

## Flexibility analysis

We used the relion_flex_analyse program to perform a principal component analysis (PCA) on the relative orientations of the bodies of a subset of 110,367 particles (Zivanov *et al*, 2018). The PCA is performed on six variables per body (3 translations and 3 rotations).

We analysed the variance in the rotations and translation of the bodies explained by the different eigenvectors. UCSF Chimera 1.15 was used to generate movies of the reconstructed body densities repositioned along these eigenvectors. Individual maps of the bodies, positioned relative to each other according to the rotations and translations corresponding to the centre of the amplitude along the different eigenvectors, were used to calculate the rotation angles and the translation distances (Pettersen *et al*, 2004).

## Model building and refinement

The *Drosophila* TRAPP core subunits were modelled using Modeller (Sali & Blundell, 1993; Webb & Sali, 2016). Previously reported crystal structures for the subunits were used as homology models (1HQ3 (Jang *et al*, 2002); 2J3T and 2J3W (Kim *et al*, 2006), 3PR6 (Wang *et al*, 2014)). The core subunit models were initially fitted into the maps using UCSF Chimera 1.15, and the chains were manually adjusted in Coot 0.9 (Pettersen *et al*, 2004; Burnley *et al*, 2017; Casañal *et al*, 2020). The final models were then refined in Phenix within the real-space refinement module, using secondary structure and Ramachandran restraints (Liebschner *et al*, 2019). The TRAPPC8 and TRAPPC11 N-terminal regions were built *de novo*. Initial models, generated using trRosetta (Yang *et al*, 2020), were docked into the corresponding map and manually adjusted in Coot 0.9 (Casañal *et al*, 2020). Regions in which the sequence could be unambiguously docked and/or supported by cross-linking data were built and kept in the final models, which were refined against the whole maps and evaluated in Phenix (Liebschner *et al*, 2019). Figures were generated using PyMOL (version 2.0 Schrödinger, LLC), UCSF Chimera 1.15 and UCSF Chimera X (Pettersen *et al*, 2004, 2020). Model geometry evaluation and half-map validation were performed using Molprobity (Williams *et al*, 2018). The final refinement statistics are provided in Table 1.

## Cross-linking coupled to mass spectrometry (XL-MS)

300 µl of TRAPPIII and TRAPPII in Buffer B at ~0.8–1 mg/ml (1.8–2 mM) were cross-linked with the N-hydroxysuccinimide (NHS) ester disuccinimidyl dibutyric urea (DSBU, formerly BuUrBu). Cross-linking was at 45 min at room temperature at 150 times the protein concentration, and then quenched by the addition of $NH_4HCO_3$ to a final concentration of 50 mM, and incubating for 15 min. The cross-linked samples were precipitated with methanol/chloroform (Wessel & Flügge, 1984), resuspended in 8 M urea, reduced with 10 mM DTT and alkylated with 50 mM iodoacetamide. Following alkylation, proteins were diluted with 50 mM $NH_4HCO_3$ to a final concentration of 2 M urea and digested with trypsin (Promega, UK), at an enzyme-to-substrate ratio of 1:20, overnight at 37°C or sequentially with trypsin and Glu-C (Promega, UK) at an enzyme-to-substrate ratio of 1:20 and 1:50 at 37°C and 25°C, respectively. The samples were acidified with formic acid to a final concentration of 2% (v/v) then split into two equal amounts for peptide fractionation by peptide size exclusion and reverse phase C18 high pH chromatography (C18-Hi-pH). For peptide size exclusion, a Superdex Peptide 3.2/300 column (GE Healthcare) with 30% (v/v) acetonitrile/0.1% (v/v) TFA as mobile phase and a flow rate of 50 µl/min was used, and fractions collected every two minutes over the elution volume of 1.0–1.7 ml. C18-Hi-pH fractionation was

carried out on an Acquity UPLC CSH C18 1.7 μm, 1.0 × 100 mm column (Waters) over a gradient of acetonitrile 2–40% (v/v) and ammonium hydrogen bicarbonate 100 mM.

The fractions were lyophilised and resuspended in 2% (v/v) acetonitrile and 2% (v/v) formic acid and analysed by nano-scale capillary LC-MS/MS using an Ultimate U3000 HPLC (Thermo Fisher Dionex, USA) to deliver a flow of approximately 300 nl/min. A C18 Acclaim PepMap100 5 μm, 100 μm × 20 mm nanoViper (Thermo Fisher Dionex, USA), trapped the peptides before separation on a C18 Acclaim PepMap100 3 μm, 75 μm × 250 mm nanoViper (Thermo Fisher Dionex, USA). Peptides were eluted with a gradient of acetonitrile. The analytical column outlet was directly interfaced via a nano-flow electrospray ionisation source, with a hybrid quadrupole orbitrap mass spectrometer (Orbitrap Q-Exactive HF-X, Thermo Scientific). MS data were acquired in data-dependent mode. High-resolution full scans (R = 120,000, $m/z$ 350–2,000) were recorded in the Orbitrap followed by higher energy collision dissociation (HCD, stepped collision energy 30 ± 3) of the 10 most intense MS peaks. MS/MS scans (R = 45,000) were acquired with a dynamic exclusion window of 20s being applied.

For data analysis, Xcalibur raw files were converted to MGF format by MSConvert (Proteowizard) and put into MeroX (Kessner *et al*, 2008; Götze *et al*, 2012). Searches were performed against an ad hoc protein database containing the sequences of the complexes and randomised decoy sequences generated by the software. The following parameters were set for the searches: a maximum number of missed cleavages of three; targeted residues K, S, Y and T; minimum peptide length of five amino acids; variable modifications: carbamidomethyl-Cys (mass shift 57.02146 Da), Met-oxidation (mass shift 15.99491 Da); DSBU modification fragments: 85.05276 Da and 111.03203 (precision: 5 ppm MS1 and 10 ppm MS2); false discovery rate cut-off: 5%. Finally, each fragmentation spectrum was manually inspected and validated. Data were analysed and figures generated using xiView (github.com/Rappsilber-Laboratory/xiView) and Xlink Analyzer (Kosinski *et al*, 2015).

### GEF activity assays

For GEF assays, the activity on His-tagged Rabs was determined by the exchange of mant-GDP for GTP using a PHERASTAR plate reader (Riedel *et al*, 2017). All Rabs and TRAPP complexes were buffer exchanged into HKM (20 mM HEPES-KOH, pH 7.4, 150 mM KOAc, 2 mM MgCl$_2$ and 1 mM DTT). Reactions containing 250 nM mant-GDP-labelled Rab alone, Rab and 200 μM GTP, or adding to the mix 10 mM EDTA, or 25 nM of the corresponding GEF, were set up in 96-well black plates (Corning), and fluorescence decay was measured at 30°C. For the comparison between the TRAPPIII and the core effect on Rab1, 50 nM GEF was used, and the assay performed at 37°C.

## Data availability

The final reconstructed maps from each frame and the weighted sum are deposited in the Electron Microscopy Data Bank (https://www.ebi.ac.uk/pdbe/emdb/) (TRAPPIII consensus map: EMD-12056, mini-TRAPPIII consensus map: EMD-12063, body 1-Core: EMD-12052, body 2-C8/C12/C13: EMD-12053, body 3-C11: EMD-12054, TRAPPII:

EMD-12066). The refined atomic models are deposited in the Protein Data Bank (http://www.wwpdb.org/) (TRAPPIII: 7B6R, MiniTRAP-PIII: 7B7O, TRAPP Core: 7B6D, C8: 7B6E, C11: 7B6H). Cross-linking mass spectrometry data are summarised in Table EV1, and mass spectrometry proteomic data have been deposited to the ProteomeX-change Consortium via the PRIDE partner repository (https://www.ebi.ac.uk/pride/), with the data set identifier PXD025064.

*Expanded View* for this article is available online.

## Acknowledgements

We thank Giuseppe Cannone, Grigory Sharov and Anna Yeates from the MRC LMB, and the eBIC Diamond staff, for assistance in cryo-EM data collection; Stephen McLaughlin and Chris Johnson for help with biophysics and Jake Grimmett and Toby Darling for computational support. We are indebted to Ester Vazquéz and Ana Casañal for advice on handling cryo-EM samples and helping in data collection, Pavel Afanasyev and Arka Chakraborty for advice on image processing. We are grateful to Andrea Nans, Tim Stevens, Elyse Fischer, Alba Herrero and Ana Torroja for help with goCTF and CryoSParc software, and to Alison Gillingham and Jérôme Cattin-Ortolá for comments on the text. This study made use of electron microscopes at the MRC LMB EM Facility and the UK's national Electron Bio-imaging Centre (eBIC) under proposal EM17434 funded by MRC. Vicente Planelles-Herrero was supported by a Long-Term Fellowship from the European Molecular Biology Organisation (EMBO). All other funding was from the Medical Research Council, as part of United Kingdom Research and Innovation (also known as UK Research and Innovation) (File reference number MC_U105178783).

## Author contributions

AG and SM conceptualised and wrote—original draft; AG investigated and visualised; AG and VJP-H involved in formal analysis; GD contributed to methodology; SM involved in funding acquisition and wrote—review and editing.

## Conflict of interest

The authors declare that they have no conflict of interest.

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
