## [Review Process File · The EMBO Journal]

Cryo-EM structure of metazoan TRAPPIII, the multi-subunit complex that activates the GTPase Rab1

Antonio Galindo, Vicente Planelles-Herrero, Gianluca Degliesposti, and Sean Munro
DOI: [10.15252/embj.2020107608](https://doi.org/10.15252/embj.2020107608)

Corresponding author(s): Sean Munro (sean@mrc-lmb.cam.ac.uk) , Antonio Galindo (agalindo@mrc-lmb.cam.ac.uk)

Review Timeline:

Submission Date:	29th Dec 20
Editorial Decision:	9th Feb 21
Revision Received:	10th Mar 21
Editorial Decision:	23rd Mar 21
Revision Received:	30th Mar 21
Accepted:	11th Apr 21

Editor: Elisabetta Argenzio

Transaction Report:

Thank you for submitting your manuscript entitled "A cryo-EM structure of metazoan TRAPPIII, the multisubunit complex that activates the GTPase Rab1" (EMBOJ-2020- 107608) to The EMBO Journal. Your study has now been assessed by three reviewers, whose reports are enclosed below for your information.

As you can see, the referees find your work interesting, but also suggest that you address a few points in order to strengthen the main conclusions.

Given the overall interest of your study, we have decided to invite you to submit a new version of the manuscript revised according to the referees' requests. I should add that it is The EMBO Journal policy to allow only a single round of revision, and acceptance of your manuscript will therefore depend on the completeness of your responses in the revised version.

We generally grant three months as standard revision time. As we are aware that many laboratories cannot function at full capacity owing to the COVID-19 pandemic, we may relax this deadline. Also, we have decided to apply our 'scooping protection policy' to the time span required for you to fully revise your manuscript and address the experimental issues highlighted herein. Nevertheless, please inform us as soon as a paper with related content is published elsewhere.

When preparing your letter of response to the referees' comments, please bear in mind that this will form part of the Review Process File and will therefore be made available online. For more details on our Transparent Editorial Process, please visit our website:
http://emboj.embopress.org/about#Transparent_Process

Before submitting your revised manuscript, deposit any primary datasets and computer code produced in this study in an appropriate public database (see <http://msb.embopress.org/authorguide#dataavailability>). Please remember to provide a reviewer password, in case such datasets are not yet public. The accession numbers and database names should be listed in a formal "Data Availability" section (placed after Materials & Method). Provide a "Data availability" section even if there are no primary datasets produced in the study.

Feel free to contact me if you have any questions about the submission of the revised manuscript to The EMBO Journal. I thank you again for the opportunity to consider this work for publication and look forward to your revision.

Referee #1:

The central result presented in this important manuscript is the cryo-EM structure of *Drosophila* TRAPPIII. TRAPPIII was originally thought to be a multisubunit tethering complex but instead appears to function primarily or exclusively as a GEF specific for Rab1, a key regulator of both the early secretory pathway and autophagy. The structure of metazoan TRAPPIII turns out to be substantially more elaborate than the structure of *S. cerevisiae* TRAPPIII, and its elucidation is a landmark achievement. The structure reveals the organization of the core subunits and two large

arms; shows that a mutation implicated in a genetic disease (spondyloepiphyseal dysplasia tarda) is located in one of the core-arm interfaces; implies that one of the arms likely contacts Rab1 and may regulate the specificity and/or activity of the GEF; and has interesting evolutionary implications. Overall, the structure is likely to be of high interest to the membrane trafficking field. Therefore, I view this manuscript as a suitable candidate for publication in EMBO J.

1. I would suggest that the authors point out explicitly that major portions of the structure - large parts of C8 and C11 as well as all of C12 and C13 - are not modeled.
2. The C2-C8 and C2L-C11 interfaces are described as being 'similar', but this claim was hard for me to evaluate. I think the authors could do more to support this point through description and illustration. For example, mightn't some sort of side-by-side comparison or overlay be helpful?
3. The reader would benefit from a more careful description of Fig. 5C, including how the illustrated orientation (of TRAPPIII with respect to the membrane) was chosen and what the relationship is between the two panels. (In regard to the two panels of Fig. 5C - does one image represent a rotation of the complex around the Z-axis with respect to the other? If so, why does the structure as shown on the right seem to extend further above the membrane than the structure as shown on the left?)
4. The manuscript would, of course, be even stronger if it included a structure of TRAPPIII bound to Rab1. Is this something the authors attempted? According to their analysis of TRAPPIII flexibility, it seems plausible that the presence of the Rab might be helpful in obtaining higher resolution, especially in regions that aren't modeled in the current structure.

Referee #2:

In this study the authors report the cryo-EM derived structures of *Drosophila* TRAPPIII and TRAPPII, although the TRAPPII structure is examined less detail. TRAPPs function as GEFs for members of the small GTPase family known as the Rabs. TRAPPIII in yeast is comprised of the TRAPP core plus an additional TRAPPIII specific component termed Trs85. By contrast metazoan TRAPPIII contains 4 TRAPPIII specific components: TRAPPC8 (Trs85) and TRAPPC11, C12 and C13 - in addition to the core associated TRAPP2CL (which is absent from yeast TRAPPIII). Remarkably, although the TRAPP-specific components are distinct to each complex the overall arrangement of TRAPPII and TRAPPIII are strikingly similar wherein both adopt a flattened triangular arrangement. Although the manuscript does not explore the cell physiological significance of the structures of TRAPPIII or TRAPPII this does not in my view limit the significance of the findings presented. The manuscript is laid out clearly and well written, and will undoubtedly be of significant interest to cell biologists with interests in the role of GTPase GEFs, membrane trafficking and in autophagy.

Specific comments:

The authors discuss technical issues related to apparent differences in 2D and 3D as well as issues related to portions of the TRAPPIII complex with poor secondary structure. As I am not an expert in protein structure determination (nor Cryo-EM) I am not in a position to critic the authors interpretations / explanations from a technical point of view. That said, the authors have gone to great lengths (via chemical crosslinking and mass spectroscopy) to valid the arrangement of the 12 TRAPPIII subunits observed in their structure (and this was also done for TRAPPII). Although the

location of Rab1 in TRAPPIII was not directly addressed, the authors were able to model the position of Rab1 within their structure of TRAPPIII.

The authors also examine the structure of *Drosophila* TRAPPII which has a similar architecture to that of yeast TRAPPII - although the yeast complex contains an additional yeast-specific subunit (Trs65). Remarkably, the overall structures of TRAPPII and TRAPPIII appear to be similar (appearing as flattened triangles) - in which the TRAPP core is held between two "arms" comprised of the TRAPP-specific subunits.

The manuscript is well written and laid out, and the discussion is thought provoking. I have no suggestions for changes to the manuscript apart from what I suggest below.

Suggestions for additions in the event of back-to-back publication with Joiner et al.

In my opinion both papers have strong merits for "separate" publication, and taken together, provide tremendous insight in the structure - function of TRAPP complexes. It would appear that the nature by which Rab1 engages TRAPPIII is slightly different to that between Ypt1 and yeast TRAPPIII, the former directly involving TRAPPC2 (Trs85) whereas the latter does not. Should these papers be published back-to-back - it would be helpful to readers if this difference could be highlighted together with some discussion of the cell biological implications of the different binding arrangements.

Referee #3:

This paper uses cryo-EM structural analysis to examine *Drosophila* TRAPPIII, the multisubunit GEF for Rab1. Resolution of the TRAPPIII structure was modest, in the range of 4-5 Å, but it was complemented by cross-linking mass spectrometry to provide convincing evidence for the arrangement of the subunits. The results highlight the locations and interactions of the four TRAPPIII-specific auxiliary subunits and clarify the effects of certain disease-causing mutations. Auxiliary subunits attach to the TRAPP core in a fashion suggesting that they aid in recruitment of TRAPPIII to specific membranes, and that they could potentially regulate TRAPPIII activity by controlling access to the active site through the movement of flexible "arms". A preliminary lower-resolution structure of the TRAPPII complex for Rab11 is also presented.

Not being an expert in structural biology, I can only comment that the analysis seems to be solid, and the findings are presented in a clear and satisfying way. Direct biological insights from this work are limited, but this type of structural information is an important advance because it establishes the foundation for specific functional studies.

A minor comment: the legend to Figure 5C is inadequate. There are words missing in this sentence: "The distance between the of the Rab1-binding site...". Also, there is no description of the right half of Figure 5C-is it a 90{degree sign} rotation?

Revisions in Response to Referees' Comments.

We are very grateful to the reviewers for their positive comments about our work and their constructive suggestions for improvements. We have followed these suggestions as described below.

Referee #1.

I would suggest that the authors point out explicitly that major portions of the structure - large parts of C8 and C11 as well as all of C12 and C13 - are not modeled.

We have added text to the results section to explicitly state this important point, as follows, *“Overall, we were able to model the core, and the N-terminal halves of TRAPPC8 and TRAPPC11. The C-terminal halves of these subunits, along with TRAPPC12 and TRAPPC13, were unmodelled as although helices were recognisable in many regions, the sequences could not be attributed. Nonetheless, the density map clearly shows the overall architecture of the entire complex.”*

2. The C2-C8 and C2L-C11 interfaces are described as being 'similar', but this claim was hard for me to evaluate. I think the authors could do more to support this point through description and illustration. For example, mightn't some sort of side-by-side comparison or overlay be helpful?

This is a helpful suggestion, and we have now added a figure showing an overlay of the C2-C8 and C2L-C11 interactions (Figure EV2). In addition, there is an overlay of the whole of C2 and C2L to show that the two subunits have a very similar overall fold (Figure EV3).

3. The reader would benefit from a more careful description of Fig. 5C, including how the illustrated orientation (of TRAPPIII with respect to the membrane) was chosen and what the relationship is between the two panels. (In regard to the two panels of Fig. 5C - does one image represent a rotation of the complex around the Z-axis with respect to the other? If so, why does the structure as shown on the right seem to extend further above the membrane than the structure as shown on the left?)

The structure on the right appeared to extend further above the membrane due to a slight perspective effect added by ChimeraX. This has now been corrected (now Fig 6A), and a rotation sign has been added between the panels to make it clear that one image does indeed represent a 90° rotation around the Z-axis with respect to the other. In addition, we have expanded the text in the results to make it clear how this putative orientation was chosen, adding the following: *“It should be stressed that this orientation on the surface is hypothetical, based on the assumption that the vertexes of the complex serve to mediate membrane contact.”*

4. The manuscript would, of course, be even stronger if it included a structure of TRAPPIII bound to Rab1. Is this something the authors attempted? According to their analysis of TRAPPIII flexibility, it seems plausible that the presence of the Rab might be helpful in obtaining higher resolution, especially in regions that aren't modeled in the current structure.

With the benefit of hindsight, we can now see that attempting a structure bound to Rab1 could well have been informative. We intend to try this now, but it is likely to take quite a long time to optimise grids and accumulate enough microscope sessions to collect sufficient particles to generate a high-resolution structure. Nonetheless, our findings will at least serve as a guide to others interested in further structural work on TRAPPIII or TRAPPII. The second mouse might get the cheese.

However, we have at least now compared the Rab1 GEF activity of the TRAPP core with that of the entire TRAPPIII complex, and found that the latter is strikingly more active. This provides good evidence that the TRAPPC8 subunit does indeed contact Rab1 when the

latter is bound to the active site, and hence TRAPPC8 contributes to the GEF activity of the complex as well as potentially increasing the specificity for Rab1 over Rab11. This data has been added to the paper as Figure 5C, and discussed in the text: "*Indeed, the entire TRAPPIII complex shows significantly more exchange activity on Rab1 than does the core alone, even when the two are compared in the absence of liposomes, consistent with the presence of TRAPPC8 promoting the interaction of Rab1 with the complex (Fig 5C).*".

Referee #2

In my opinion both papers have strong merits for "separate" publication, and taken together, provide tremendous insight in the structure - function of TRAPP complexes. It would appear that the nature by which Rab1 engages TRAPPIII is slightly different to that between Ypt1 and yeast TRAPPIII, the former directly involving TRAPPC2 (Trs85) whereas the later does not. Should these papers be published back-to-back - it would be helpful to readers if this difference could be highlighted together with some discussion of the cell biological implications of the different binding arrangements.

To aid comparison of the TRAPPIII complexes in metazoans and yeast we have added a comparison of their overall structures which illustrates the important point raised by the reviewer that direct contact between Rab1 and TRAPPC8 is possible in metazoans, but not possible for their yeast orthologues (Figure 6B). We now mention this in the Discussion and briefly discuss the biological implications. "*Modelling of Rab1 into the complex, and comparison of the GEF activity of the whole complex with the core, both suggest that Rab1 contacts the TRAPPC8 arm positioned above the active site. This part of TRAPPC8 is present in most species from humans to plants and protozoa, but has been lost in some fungi. Thus, in S. cerevisiae, Trs85 corresponds to the first ~650 residues of the 1319 residue Drosophila TRAPPC8. This presumably reflects there being no TRAPPC11, TRAPPC12 and TRAPPC13 to connect to. Indeed, the Drosophila TRAPPIII core plus the first half of TRAPPC8 fits closely into the overall shape of S. cerevisiae TRAPPIII determined by negative stain EM (Fig 6B) (Tan et al, 2013). The TRAPP core in yeast is sufficient to activate the Rab1 orthologue Ypt1 in solution, with the presence of Trs85 increasing the activity toward the GTPase when bound to membranes, presumably by promoting membrane recruitment of the complex rather than via direct binding of Trs85 to Ypt1 (Thomas et al, 2017).*"

Referee #3

A minor comment: the legend to Figure 5C is inadequate. There are words missing in this sentence: "The distance between the of the Rab1-binding site... ". Also, there is no description of the right half of Figure 5C-is it a 90{degree sign} rotation?

We apologise for this error, and it has been corrected in the legend of what is now Figure 6A. We have also added a rotation symbol between the two halves of the figure to make it clear that the right view is indeed a 90° rotation of the one on the left.

1st Revision - Editorial Decision**23rd Mar 2021**

Thank you for submitting your revised study. I have now checked your manuscript and the point-by-point rebuttal letter and find that the referees' points have been sufficiently addressed.

However, there are few editorial issues concerning the text and the figures that I need you to address before we can officially accept your manuscript.

2nd Revision - Editorial Decision**11th Apr 2021**

I am pleased to inform you that your manuscript has been accepted for publication in The EMBO Journal.

Corresponding Author Name: Sean Munro

Manuscript Number: EMBOJ-2020-107608